# Hydrogenosomal tail-anchored proteins are targeted to both mitochondria and ER upon their expression in yeast cells

**Andrè Ferdigg**[ID], **Kai S. Dimmer, Doron Rapaport** *, **Daniela G. Vitali**¤*

Interfaculty Institute of Biochemistry, University of Tübingen, Tübingen, Germany

¤ Current address: Sir William Dunn School of Pathology, University of Oxford, Oxford, United Kingdom
* doron.rapaport@uni-tuebingen.de (DR); daniela.vitali@path.ox.ac.uk (DGV)

**Data Availability Statement:** All relevant data are within the manuscript and its Supporting Information files.

## Abstract

Some organisms, like *Trichomonas vaginalis*, contain mitochondria-related hydrogen-producing organelles, called hydrogenosomes. The protein targeting into these organelles is proposed to be similar to the well-studied mitochondria import. Indeed, *S. cerevisiae* mitochondria and *T. vaginalis* hydrogenosomes share some components of protein import complexes. However, it is still unknown whether targeting signals directing substrate proteins to hydrogenosomes can support in other eukaryotes specific mitochondrial localization. To address this issue, we investigated the intracellular localization of three hydrogenosomal tail-anchored proteins expressed in yeast cells. We observed that these proteins were targeted to both mitochondria and ER with a variable dependency on the mitochondrial MIM complex. Our results suggest that the targeting signal of TA proteins are only partially conserved between hydrogenosomes and yeast mitochondria.

## Introduction

Mitochondria are essential organelles that evolved from the endosymbiosis of α-proteobacteria and ancestral archaea cells [1]. One of their main function is to produce ATP using oxygen as final electron acceptor. Under oxygen-restricted conditions, these organelles evolved to mitochondria-related organelles, such as hydrogenosomes, which produce ATP and hydrogen via fermentation [2, 3]. The most studied organism containing hydrogenosomes is the parasitic protist *Trichomonas vaginalis* that inhabits the human urogenital tract [4]. Like mitochondria, hydrogenosomes are surrounded by two membranes and import proteins post-translationally from the cytosol [5]. Interestingly, some key components of the mitochondria import machineries, such as subunits of the translocases of the outer and inner membranes (TOM and TIM complexes, respectively), are conserved between *S. cerevisiae* and *T. vaginalis*, although the two organisms belong to evolutionary very distant supergroups (Opistokonta and Excavata, respectively) [6, 7]. This observation underlies the relation in the evolution of both organelles [2, 7]. Despite such similarities, it is not clear whether the protein targeting signals are conserved between mitochondria and hydrogenosomes.

**Funding:** This work was supported by the Deutsche Forschungsgemeinschaft (RA 1028/7-2 and RA 1028/10-1 to D.R.).

**Competing interests:** The authors have declared that no competing interests exist.

Proteomic analysis identified 70 putative hydrogenosomal membrane proteins in *T. vaginalis*, including 12 putative tail-anchored (TA) proteins [7, 8]. TA proteins are single span membrane proteins with a transmembrane segment (TMS) localized close to the C-terminus of the polypeptide. These proteins can be inserted into ER, mitochondrial, or peroxisomal membrane, and expose the large soluble domain towards the cytosol. Since the TMS, which contains the targeting information, is the last portion of the polypeptide to be released from the ribosome, TA proteins can be imported only post-translationally [9]. The physicochemical properties of the TMS and its flanking regions seem to dictate the target organelle for these proteins. ER TA proteins generally have longer and more hydrophobic TMS, while mitochondrial proteins are characterized by shorter and less hydrophobic TMS, flanked by positively charged residues on both sides. Finally, low hydrophobicity of the TMS, coupled with basic residues at the C-terminal element (CTE) directs TA proteins to peroxisomes [9, 10]. An additional feature, which seems to influence the targeting of TA proteins, is the helical content. The TMSs of mitochondrial and peroxisomal TA proteins tend to have lower helical content than those targeted to ER [11].

TA proteins follow different pathways of integration into the mitochondrial outer membrane (MOM) [12–14]. Fis1, for example, does not require any of the known import machineries and seems to be inserted in an unassisted manner that depends on the membrane lipid composition [14–16]. On the other hand, other TA proteins, like Gem1, Tom6, or Tom7 follow a route that partially involves the MIM complex [15, 17].

To study the conservation of the targeting information of TA proteins between hydrogenosomes and mitochondria, we expressed three *T. vaginalis* TA proteins in *S. cerevisiae* cells and assessed their subcellular localization. We observed that the hydrogenosomal TA proteins did not target only to mitochondria but also, to different extents, to the ER and peroxisomes. These findings suggest that the targeting signal of such proteins evolved independently in these two organelles.

## Materials and methods

### Yeast strains and growth conditions

Yeast strains used in this study were isogenic to *Saccharomyces cerevisiae* strains W303a/α or YPH499. Standard genetic techniques were used for growth and manipulation of yeast strains. Yeast cells were grown in synthetic medium S (0.67% [w/v] bacto-yeast nitrogen base without amino acids) with glucose (2% [w/v], D), galactose (2% [w/v], Gal), sucrose (2% [w/v], Suc), or glycerol (3% [w/v], G) as carbon source. Transformation of yeast cells was performed by the lithium acetate method. Table 1 includes a list of strains used in this study.

**Table 1. Yeast strains used in this study.**

| Name | Mating type | Genetic background | Source or reference |
|------|-------------|--------------------|--------------------|
| *fis1Δ* | *MATa* | YPH499, *ura3-52 lys2-801_amber ade2-101 _ochre trp1-D63 his3-D200 leu2-D1, fis1Δ::His3* | [18] |
| W303a | *MATa* | *ade2-1 can1-100 his3-11 leu2 3_112 trp1Δ2 ura3-52* | Lab stock |
| *mim1Δ* | *MATa* | W303a, *mim1Δ::His3* | [19] |
| W303α | *MATα* | *ade2-1 can1-100 his3-11 leu2 3_112 trp1Δ2 ura3-52* | Lab stock |
| *gem1Δ* | *MATα* | W303α, *gem1Δ::His3* | This study |

## Drop dilution assay

Cells were grown in a liquid selective medium to logarithmic phase, harvested, resuspended in water to $OD_{600} = 1$ and diluted fivefold in water. Subsequently, 5 µl of each cell suspension were spotted on solid media and the plates were incubated at the indicated temperature for 3 to 8 days.

## Recombinant DNA techniques

The cDNAs of TA4 and TA10 were amplified by PCR with primers containing NcoI and SalI restriction sites, using pVagTag-2HA-TA4 and pVagTag-2HA-TA10 [8], respectively, as templates. The obtained DNA fragments were inserted downstream of the 3xHA tag into the yeast expression vector pYX142, digested with the same restriction enzymes. To construct the pYX142-3HA-TA7 plasmid, the coding sequence for TA7 was amplified by PCR from pTag-Vag-2HA-TA7 [8], using primers containing the restriction sites for EcoRI and NheI. This DNA fragment was inserted downstream of the 3xHA tag of the plasmid pRS316-*FIS1*pr-3HA-Fis1(cyt)-*FIS1ter* [15], which was digested with the same restriction enzymes. Finally, the DNA fragment coding for 3xHA-TA7 was subcloned into the pYX142 vector using XmaI and NheI restriction sites. The plasmids pYX142-eGFP-TA4, pYX142-eGFP-TA7, and pYX142--eGFP-TA10 were obtained by amplifying the coding sequences of TA4, TA7, and TA10 from pYX142-3HA-TA4, pYX142-3HA-TA7, and pYX142-3HA-TA10, respectively, with primers containing BamHI and SalI restriction sites, and inserting them into the pYX142-eGFP plasmid, downstream the eGFP coding sequence.

The plasmids pRS316-*FIS1*pr-3HA-Gem1(cyt)-TA7(TMS)-*FIS1*ter and pRS316-*FIS1*pr-3HA-Gem1(cyt)-TA10(TMS)-*FIS1*ter were obtained by replacing the sequence coding for Fis1 (TMS) of the pRS316-*FIS1*pr-3HA-Gem1(cyt)-Fis1(TMS)-*FIS1*ter [15] with the sequences encoding for the TMS, including part of the N-terminal element (NTE) and the whole CTE, of either TA7 or TA10 amplified from the plasmids pYX142-3HA-TA7 and pYX142-3HA-TA10, respectively. Primers containing SalI and HindIII restriction sites were used. To clone the pRS316-*FIS1*pr-3HA-Gem1(cyt)-TA4(TMS)-*FIS1*ter plasmid, the sequence encoding the TMS and its flanking regions was amplified from the plasmid pYX142-3HA-TA4, with primers harbouring SalI and XhoI restriction sites and inserted in-frame into the pRS316-*FIS1*pr-3HA-Gem1(cyt) plasmid lacking the STOP codon. Subsequently, the Fis1 terminator was amplified by PCR from pRS316-*FIS1*pr-3HA-Gem1(cyt)-Fis1(TMS)-*FIS1*ter [15] with primers containing XhoI and KpnI restriction sites and inserted into the pRS316-*FIS1*pr-3HA-Gem1 (cyt)-TA4(TMS) plasmid.

The plasmid pRS316-*FIS1*pr-3HA-Gem1(cyt)-*FIS1*ter was obtained by amplifying from the pRS316-*FIS1*pr-3HA-Gem1(cyt)-TA4(TMS)-*FIS1*ter plasmid the sequence corresponding to the portion of Gem1(cyt), downstream the NheI cutting site (present in *GEM1* gene), with primers containing NheI and HindIII restriction sites. To remove the sequence encoding the TA4(TMS) sequence, the PCR product was inserted into the pRS316-*FIS1*pr-3HA-Gem1(cyt)-TA4(TMS)-*FIS1*ter plasmid, digested with the same restriction enzymes. Tables 2 and 3 contain lists of the plasmids and primers, respectively, used in this study.

## Subcellular fractionation

Isolation of mitochondria from yeast cells was performed by differential centrifugation, as previously described [20]. To obtain highly pure mitochondria, isolated organelles were layered on top of a Percoll gradient and isolated according to a published procedure [21]. The post-mitochondrial fraction was clarified at 18000 x g for 15 min at 2˚C and subsequently subjected to ultracentrifugation at 200000 x g for 1 h at 2˚C. A sample of the supernatant was collected as

**Table 2. Plasmids used in this study.**

| Plasmids | Promoter | Coding sequence (aa) | Markers | Source or reference |
|---|---|---|---|---|
| pYX142-3HA-TA4 | TPI | 3HA-TA4 full lengh | LEU2, Amp$^R$ | This study |
| pYX142-3HA-TA7 | TPI | 3HA-TA7 full lengh | LEU2, Amp$^R$ | This study |
| pYX142-3HA-TA10 | TPI | 3HA-TA10 full lengh | LEU2, Amp$^R$ | This study |
| pYX132-Fis1(TMC) | TPI | Fis1 with a single Cys in the TMS | TRP1, Amp$^R$ | [16] |
| pYX142-eGFP-TA4 | TPI | GFP-TA4 full lengh | LEU2, Amp$^R$ | This study |
| pYX142-eGFP-TA7 | TPI | GFP-TA7 full lengh | LEU2, Amp$^R$ | This study |
| pYX142-eGFP-TA10 | TPI | GFP-TA10 full lengh | LEU2, Amp$^R$ | This study |
| pRS426-mt-RFP | TPI | Mt-RFP | URA3, Amp$^R$ | Lab stock |
| pRS416-HDEL-dsRed | - | HDEL-dsRed | URA3, Amp$^R$ | Lab of Maya Schuldiner |
| RFP-PTS1 | - | RFP-PTS1 | URA3, Amp$^R$ | Lab of Maya Schuldiner |
| pRS316-3HA-Gem1 | FIS1pr | 3HA-Gem1 full lengh | URA3, Amp$^R$ | [15] |
| pRS316-3HA-Gem1(cyt)-TA4(TMS) | FIS1pr | 3HA-Gem1(1–634) + TA4(393–434) | URA3, Amp$^R$ | This study |
| pRS316-3HA-Gem1(cyt)-TA7(TMS) | FIS1pr | 3HA-Gem1(1–634) + TA7(288–323) | URA3, Amp$^R$ | This study |
| pRS316-3HA-Gem1(cyt)-TA10(TMS) | FIS1pr | 3HA-Gem1(1–634) + TA10(156–188) | URA3, Amp$^R$ | This study |
| pRS316-3HA-Gem1(cyt) | FIS1pr | 3HA-Gem1(1–634) | URA3, Amp$^R$ | This study |

cytosolic fraction. To isolate the microsomes, the pellet was resuspended in 3 ml of SEM buffer (250 mM sucrose, 1 mM EDTA, 10 mM MOPS, pH 7.2) with 2 mM phenylmethylsulfonyl fluoride (PMSF), homogenised with a douncer homogenizer and clarified at 18000 x g for 20 min at 4°C. Fractions corresponding to the cytosol, microsomes, and the whole cell lysate were

**Table 3. Primers used in this study.**

| Primer name | Sequence (5'-3') | Note |
|---|---|---|
| NcoITA4F | GGGGGGCCATGGATGGATATTGAGCATTCCGTGC | Amplification of *TA4*, NcoI restriction site at 5' |
| BamHITA4F | GGGGGGGGATCCATGGATATTGAGCATTCCGTGC | Amplification of *TA4*, BamHI restriction site at 5' |
| SalITA4R | CCCCCGTCGACTTATTCTTTCTTCTTTGCTACTTTTGTG | Amplification of *TA4*, SalI restriction site at 5' |
| EcoRITA7F | GGGGGGAATTCATGGAGAACGCATTTTTGATGACTC | Amplification of *TA7*, EcoRI restriction site at 5' |
| NheITA7R | CCCCCGCTAGCTTATTTATGATTCATGAAGCGCTTAATACC | Amplification of *TA7*, NheI restriction site at 5' |
| BamHITA7F | GGGGGGGGATCCATGGAGAACGCATTTTTGATGACTC | Amplification of *TA7*, BamHI restriction site at 5' |
| SalITA7F | GGGGGGTCGACATGGAGAACGCATTTTTGATGACTC | Amplification of *TA7*, SalI restriction site at 5' |
| NcoITA10F | GGGGGCCATGGATGGAAGCTGCTAAAGAGGCTG | Amplification of *TA10*, NcoI restriction site at 5' |
| BamHITA10F | GGGGGGGGATCCATGGAAGCTGCTAAAGAGGCTG | Amplification of *TA10*, BamHI restriction site at 5' |
| SalITA10R | CCCCCGTCGACTTATTTTTTCTCATCTTTGCAGAATATGATAAC | Amplification of *TA10*, SalI restriction site at 5' |
| EcoRIGFPF2 | GGGGGGAATTCATGAGTAAGGGTGAAGAAC | Amplification of *eGFP*, EcoRI restriction site at 5' |
| BamHIGFPR | CCCCCGGATCCTTTGTATAGTTCATCCATGC | Amplification of *eGFP*, BamHI restriction site at 5' |
| SalITA4TMDF | GGGGGGTCGACTACAGACAAACGGCTTACGGTGAACCTAAGCCAAC | Amplification of TMD of TA4, SalI restriction site at 5' |
| XhoITA4R | CCCCCCTCGAGTTATTCTTTCTTCTTTGCTACTTTTGTG | Amplification of TMD of TA4, XhoI restriction site at 5' |
| XhoIFis1TerF | GGGGGCTCGAGATAAAAAATCAGCACATACGTACATACATAAGAATG | Amplification of Fis1 terminator, XhoI restriction site at 5' |
| KpnIFis1TerR | CCCCCGGTACCATCTCACAATACAGTATTACGATTTAACAATAGACTATTG | Amplification of Fis1 terminator, KpnI restriction site at 5' |
| SalITA7TMDF | GGGGGGTCGACTACAGACAAACGGCTAATCTTTCTAAAGTTGGAATTTC | Amplification of TMD of TA7, SalI restriction site at 5' |
| HindIIITA7R | CCCCCAAGCTTTTATTTATGATTCATGAAGCGCTTAATACC | Amplification of TMD of TA7, HindIII restriction site at 5' |
| SalITA410MDF | GGGGGGTCGACTACAGACAAACGGCTCAAGAAAAACTTAACTCATTC | Amplification of TMD of TA10, SalI restriction site at 5' |
| HindIIITA10R | CCCCCAAGCTTTTATTTTTTCTCATCTTTGCAGAATATGATAAC | Amplification of TMD of TA10, HindIII restriction site at 5' |
| Gem1for2NheI | GGGGGCTAGCACAATGGAGTATGACG | Amplification of Gem1(cyt), NheI restriction site at 5' |
| Gem1rev2 | GGGAAGCTTTCAAGCCGTTTGTCTGTAGTCGAC | Amplification of TMD of TA10, HindIII restriction site at 5' |

precipitated with chloroform and methanol and resuspended in 2x sample buffer, heated for 10 min at 95˚C, and analysed by SDS-PAGE and immunoblotting.

## Crude mitochondria preparation

Cells were grown in liquid media to logarithmic phase, harvested, and resuspended in SEM buffer with 2 mM PMSF. The cells were ruptured with glass beads (Ø 0.25–0.5 mm) using FastPrep-24 5G (MP Biomedicals) for 40 sec, 6.0 m/sec. To remove glass beads and cell debris, the samples were clarified (1000 x g, 3 min, 4˚C). The supernatant was then centrifuged (13200 x g, 10 min, 4˚C) and the pellet, corresponding to the crude mitochondrial fraction, was resuspended in a 2x sample solution, heated at 95˚C for 10 min, and analysed by SDS-PAGE and immunoblotting.

In other cases, like for the fractionations described in Fig 4B, cells were grown to logarithmic phase, harvested, and resuspended in resuspension buffer (100 mM Tris, 10 mM DTT). After harvesting the cells, they were resuspended in spheroblasting buffer (1.2 M Sorbitol, 20 mM KPI, pH 7.2) containing zymolyase (4.5 g/ml) and incubated at 30˚C for 1 h. Spheroblasted cells were then resuspended in homogenization buffer (0.6 M Sorbitol, 10 mM Tris pH 7.4, 1 mM EDTA, 0.2% fatty acid free BSA, 1 mM PMSF, 1x cOmplete protease inhibitor) and homogenized with a douncer. After a clarifying spin (600 x g, 5 min, 4˚C), crude mitochondria were isolated by centrifugation (18000 x g, 10 min, 4˚C). Whole cell lysate and cytosolic fractions were precipitated with chloroform and methanol, resuspended with 2x sample buffer, heated for 10 min at 95˚C, and analysed by SDS-PAGE and immunoblotting.

## Western blotting and immunodecoration

Protein samples for immunodecoration were analyzed by either 10% or 12.5% SDS-PAGE and subsequently transferred onto nitrocellulose membranes by semi-dry Western blotting. Proteins were detected by blocking the membrane with 5% skim milk and subsequently incubating them first with primary antibodies and then with horseradish peroxidase-conjugates of goat anti-rabbit or goat anti-rat secondary antibodies. Band intensities were quantified with the AIDA software (Raytest). Table 4 contains a list of the antibodies used in this study.

**Table 4. Antibodies used in this study.**

| Antibodies | Dilution | Source |
|---|---|---|
| polyclonal rat anti-HA | 1 : 1000 | 11867423001 (Roche) |
| polyclonal rabbit anti-Fis1 | 1 : 1000 | Lab stocks |
| polyclonal rabbit anti-Bmh1 | 1 : 1500 | Lab stocks |
| polyclonal rabbit anti-Erv2 | 1 : 2000 | Lab of Roland Lill |
| polyclonal rabbit anti-Pdi1 | 1 : 3000 | Lab of Blanche Schwappach |
| polyclonal rabbit anti-Tom20 | 1 : 5000 | Lab stocks |
| polyclonal rabbit anti-Aco1 | 1 : 7000 | Lab stocks |
| polyclonal rabbit anti-Mcr1 | 1 : 2000 | Lab of Carla Koehler |
| polyclonal rabbit anti-Tom70 | 1 : 5000 | Lab stocks |
| polyclonal rabbit anti-Hep1 | 1 : 3000 | Lab of Kai Hell |
| polyclonal rabbit anti-Por1 | 1 : 3000 | Lab stocks |
| polyclonal rabbit anti-Tom40 | 1 : 4000 | Lab stocks |
| Horseradish peroxidase coupled goat-ant-rabbit | 1 : 10000 | 1721019 (Bio-rad) |
| Horseradish peroxidase coupled goat-ant-rat | 1 : 3000 | 1721011 (Bio-rad) |

## Protease protection assay

A sample (50 or 100 μg) of either mitochondria or microsomes was resuspended in 100 μl of SEM buffer in the presence or absence of 1% Triton X-100. The samples were supplemented with Proteinase K (50 μg/ml) and incubated on ice for 30 min. The proteolytic reaction was stopped with 5 mM PMSF. The samples were then precipitated with trichloroacetic acid (TCA) and resuspended in 40 μl of 2x sample buffer, heated for 10 min at 95˚C, and analyzed by SDS-PAGE and immunoblotting.

## Alkaline extraction

Mitochondria or microsomes fractions (50 or 100 μg) were resuspended in 100 μl of buffer containing 10 mM HEPES-KOH, 100 mM $Na_2CO_3$, pH 11.5, and incubated for 30 min on ice. The membrane fraction was pelleted by centrifugation (76000 x g, 30 min, 2˚C) and the supernatant was precipitated with TCA. Both fractions were resuspended in 40 μl of 2x sample buffer, heated for 10 min at 95˚C, and analysed by SDS-PAGE and immunoblotting.

## Glycosylation assay

To test for glycosylation of proteins, 50 or 100 μg of microsomes fraction were resuspended in 10 μl glycoprotein denaturing buffer (0.5% SDS, 40 mM DTT) and incubated for 10 min at 95˚C. Then, the samples were supplemented with 500 units of either Endoglycosidase H (EndoH) or Peptide:N-Glycosidase F (PNGase) (New England BioLabs) in the respective buffer (according to the manufacturer's instructions) and incubated for 1 h at 37˚C. At the end of the incubation period, the samples were precipitated with TCA, resuspended in 40 μl of 2x sample buffer, heated for 10 min at 95˚C, and analysed by SDS-PAGE and immunoblotting.

## Fluorescence microscopy

Yeast cultures were grown to logarithmic phase in galactose-containing media and mixed on a glass slide in a 1:1 (v/v) ratio with 1% low melting point agarose. Fluorescence images were acquired with spinning disk microscope Zeiss Axio Examiner Z1 equipped with a CSU-X1 real-time confocal system (Visitron), VS-Laser system, and SPOT Flex CCD camera (Visitron Systems). Images were analysed with VisiView software (Visitron).

# Results and discussion

## In yeast cell TA4 localizes to the ER with an inverted topology

Recently, several putative tail-anchored proteins residing in the hydrogenosomes of *T. vaginalis* were identified [8]. To study the conservation of the targeting of TA proteins between *T. vaginalis* and *S. cerevisiae*, we expressed in yeast cells out of the five newly identified hydrogenosomal proteins the three proteins that have less hydrophobic TMS, a typical characteristic of mitochondrial TA proteins, and investigated their localization and topology. Since we do not have antibodies against these proteins, we N-terminally tagged them with three hemagglutinin epitopes (3xHA).

First, we analysed TA4 and either a plasmid expressing this protein, or an empty plasmid as a control, were transformed into yeast wild-type (WT) cells. Surprisingly, when the subcellular localization of TA4 was monitored, we observed that this protein was mainly enriched in the ER (microsomes) fraction and only a minor portion was found in mitochondria (Fig 1A). Absence of a signal for cells harbouring an empty plasmid confirmed specificity. To confirm the successful separation of the cellular compartments, the mitochondrial TA protein Fis1, the

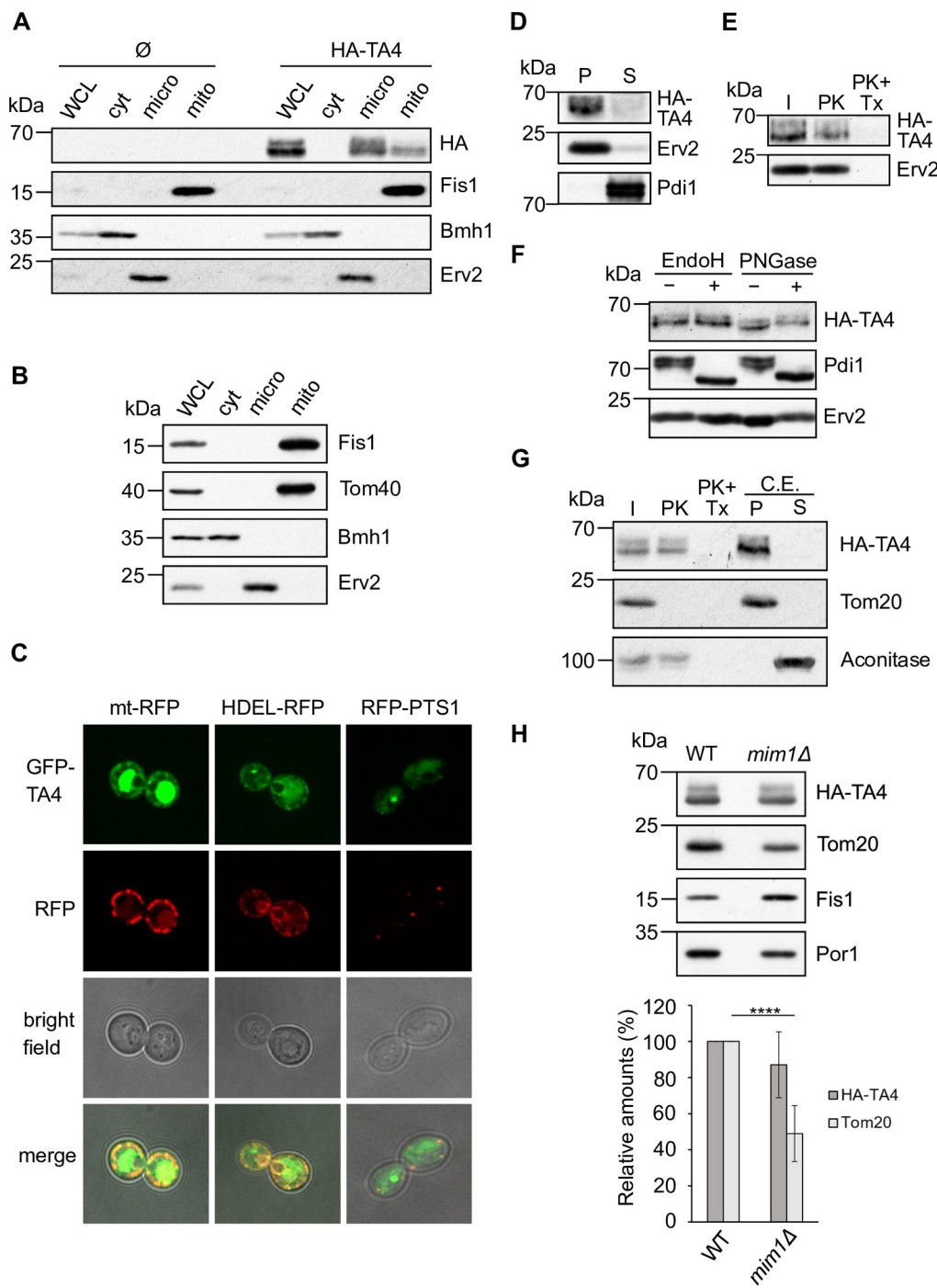

**Fig 1. TA4 is distributed between mitochondria and ER upon expression in yeast cells.** (A) Fractions corresponding to whole cell lysate (WCL), cytosol (cyt), microsomes (micro), and mitochondria (mito) were isolated from WT cells transformed with either an empty plasmid (Ø) or a plasmid expressing 3HA-TA4. The same amount of each fraction was analysed by Western blot and immunodecoration with the indicated antibodies. The mitochondrial protein Fis1, the cytosolic protein Bmh1, and the ER protein Erv2 were used as markers. (B) WCL, cytosol, microsomes and mitochondria fractions were isolated from *fis1Δ* cells overexpressing Fis1(TMC), analysed by Western blot, and immunodecorated with the indicated antibodies. Tom40 is a mitochondrial marker. (C) Representative fluorescence microscopy images of WT cells co-transformed with plasmids encoding GFP-TA4 and a plasmid encoding mitochondrial marker (mt-RFP), ER marker (HDEL-RFP), or peroxisomal one (RFP-PTS1). (D) Alkaline extraction of microsomal fraction isolated from WT cells expressing 3HA-TA4. Pellet (P) and supernatant (S) fractions were analysed by Western botting and immunodecoration with the indicated antibodies. Erv2, ER membrane protein; Pdi1, soluble ER

protein. (E) Microsomes isolated from WT cells were treated with PK in the presence or absence of TritonX-100 (TX) and analysed by Western blot and immunodecoration with the indicated antibodies. (F) Microsomes isolated from cells expressing 3HA-TA4 were treated with the enzymes EndoH or PNGase and analysed by Western blot and immunodecoration. The glycosylated protein Pdi1 was used as a control. (G) Mitochondria isolated as in (A) were treated with PK as described in (E) or were subjected to alkaline extraction as described in (D). Samples were analysed by Western blot and immunodecoration with the indicated antibodies. Tom20 is a MOM protein facing the cytosol, while Aconitase is a soluble matrix protein. (H) Top panel. Crude mitochondria isolated from WT and *mim1Δ* cells expressing 3HA-TA4 were analysed by Western blot. The MOM proteins Tom20, Fis1, and Por1 were used as controls. Tom20 is a known MIM substrate. Bottom panel. The steady state levels of 3HA-TA4 and Tom20 were quantified, normalized with PonceauS staining, and presented as percentage of their levels in WT cells. The values represent the average of at least three independent experiments and error bars the standard deviation (SD). **** P ≤ 0.0001.

ER protein Erv2, and the cytosolic protein Bmh1, were detected, as expected, in their respective fractions.

Given that the hydrogenosomal proteins were expressed under a strong promoter (TPI*pr*), it is possible that their ER localization is caused by the high protein levels. To exclude this possibility, we expressed under the same promoter the mitochondrial TA protein Fis1 in a strain deleted for the endogenous protein (*fis1Δ*). As shown in Fig 1B, despite its overexpression, Fis1 is found solely in the mitochondrial fraction suggesting that the ER location of the hydrogenosomal protein TA4 cannot be explained solely by its high expression levels.

Detecting the subcellular localization of GFP-tagged TA4 by fluorescence microscopy confirmed that the protein was targeted to ER and mitochondria, as shown by co-localization with an ER (HDEL-RFP) and a mitochondrial targeted RFP (mt-RFP) (Fig 1C). Although TA proteins can be targeted also to peroxisomes, the absence of co-localization with a peroxisomal targeted RFP (RFP-PTS1) indicates that TA4 is not targeted to these organelles (Fig 1C). Additionally, we observed a large amount of the GFP signal in the vacuole, suggesting that a subpopulation of the protein was degraded in the vacuole. We concluded that, in yeast cells, HA-TA4 is mainly localized to the ER and only partially to mitochondria, contrary to the situation in *T. vaginalis* where it was found exclusively in hydrogenosomes [8]. This implies that the targeting signals and their decoding are not completely conserved between *T. vaginalis* and *S. cerevisiae*.

Subsequently, we investigated the topology of HA-TA4 in the ER. Employing alkaline extraction of microsomes isolated from cells expressing HA-TA4, we could confirm that the protein is embedded in the ER membrane. TA4 was detected in the pellet, as the membrane protein Erv2, while the soluble protein Pdi1 was found in the supernatant (Fig 1D). Additionally, upon treatment of isolated microsomes with proteinase K (PK), we noted that the protein is protected, similar to the ER luminal protein Erv2 (Fig 1E). The addition of the detergent Triton-X to the reaction allowed the protease to degrade both proteins, confirming that they were protected by the membrane. These observations strongly suggested that TA4 is inserted into the ER membrane such that its soluble domain is exposed to the ER lumen. This phenomenon is not surprising, since we observed previously that heterologous mitochondrial TA proteins, like the rat cytochrome b5-RR, expressed in yeast cells can be partially localised to the ER, with an inverted topology [22].

Interestingly, we detected a higher molecular weight species of HA-TA4, mostly in the ER fraction (Fig 1A). We hypothesized that this slower migrating species could result from post-translational modification, possibly glycosylation. To test this idea, microsomes harbouring HA-TA4 were treated with Endoglycosidase H (EndoH) or Peptide:N-Glycosidase F (PNGase). As expected, we observed a shift in the migration of the known glycosylated protein Pdi1, while Erv2 was not affected by the treatment (Fig 1F). Interestingly, both bands corresponding to HA-TA4 were detected also after the enzymatic treatment, indicating that this

protein is not glycosylated and the shift in the migration is probably the consequence of another posttranslational modification.

Next, we investigated the topology of TA4 in the mitochondrial fraction. Also in this case, the HA signal was not affected by PK treatment, unless Triton-X was present, and the protein was found in the pellet fraction of the alkaline extraction assay (Fig 1G). As controls, we used the MOM protein Tom20, which is exposed to the cytosol and the matrix soluble protein Aconitase. These results suggest that in yeast cells, HA-TA4 assumes in both ER and mitochondria a topology with the N-terminal soluble domain facing the lumen or the IMS, respectively.

Recently, it has been observed that the MIM complex is required for the insertion of single-span proteins, including the TA protein Gem1, into the MOM [15, 17]. Therefore, we tested whether the deletion of Mim1, the central component of the MIM complex, affects the mitochondrial levels of HA-TA4. We isolated crude mitochondria from either WT or *mim1Δ* cells expressing HA-TA4 and analysed the steady state levels of this protein. As expected, the levels of the known MIM-substrate Tom20 were significantly reduced in the deletion strain, while Fis1 (a MIM-independent TA protein) was not affected (Fig 1H). Interestingly, the steady state levels of HA-TA4 were unaltered in the absence of Mim1, suggesting that the small portion of this hydrogenosomal TA protein, which is inserted into the MOM, does not require the assistance of the MIM complex. Indeed, it is possible for TA proteins to insert into the MOM in an unassisted way. A well-studied example is Fis1, which does not require any know import machinery and its targeting depends on the ergosterol content of the membrane [14–16].

In conclusion, TA4 is recognized in yeast cells mainly as an ER protein, while it can insert also into the MOM in a MIM-independent way. Notably, TA4 inserts with an inverted topology into the membrane of both organelles.

## TA7 behaves like a mitochondrial TA protein

We then turned to TA7 and observed by subcellular fractionation that once expressed in yeast cells it is found mainly in mitochondria, with a minor fraction in the ER (Fig 2A). As controls for the purity of the isolated fractions, we used the mitochondrial protein Tom70, the ER protein Erv2, and the cytosolic protein Hexokinase. The localization of TA7 in yeast mitochondria was confirmed by fluorescence microscopy and co-localization of a GFP-tagged version of the protein with mt-RFP but not with HDEL-RFP (Fig 2B). Interestingly, TA7 was also found in peroxisomes (Fig 2B). These observations suggest that the targeting signal of the hydrogenosomal protein TA7 is recognized by mitochondrial and peroxisomal targeting machineries, while it is not a substrate of the ER system. The dual targeting of TA proteins to mitochondria and peroxisomes has been observed previously for the endogenous yeast proteins Fis1 and Gem1 [23] and is not surprising, since the targeting information for both organelles is a short, moderately hydrophobic TMS flanked by positive charges. Indeed, TA7 has a TMS with low hydrophobicity, moderate helical content, and positively charged CTE (Table 5). These characteristics explain the dual targeting of this protein to both mitochondria and peroxisomes.

Subsequently, we investigated the topology of this protein in mitochondria and ER. Alkaline extraction of the microsomal fraction confirmed that HA-TA7, as Erv2, is a membrane associated protein, while the soluble protein Pdi1 was found in the soluble fraction (Fig 2C). Additionally, when we treated the ER fraction with PK, we could observe that the HA signal completely disappeared, indicating that the N-terminal domain of TA7 is facing the cytosol, as in a canonical TA protein (Fig 2D). As expected, Erv2 was protease-resistant and its band disappeared only after treatment with Triton-X and PK. Analogous experiments with isolated mitochondria revealed that also in this organelle HA-TA7 adopts a classical TA topology. It

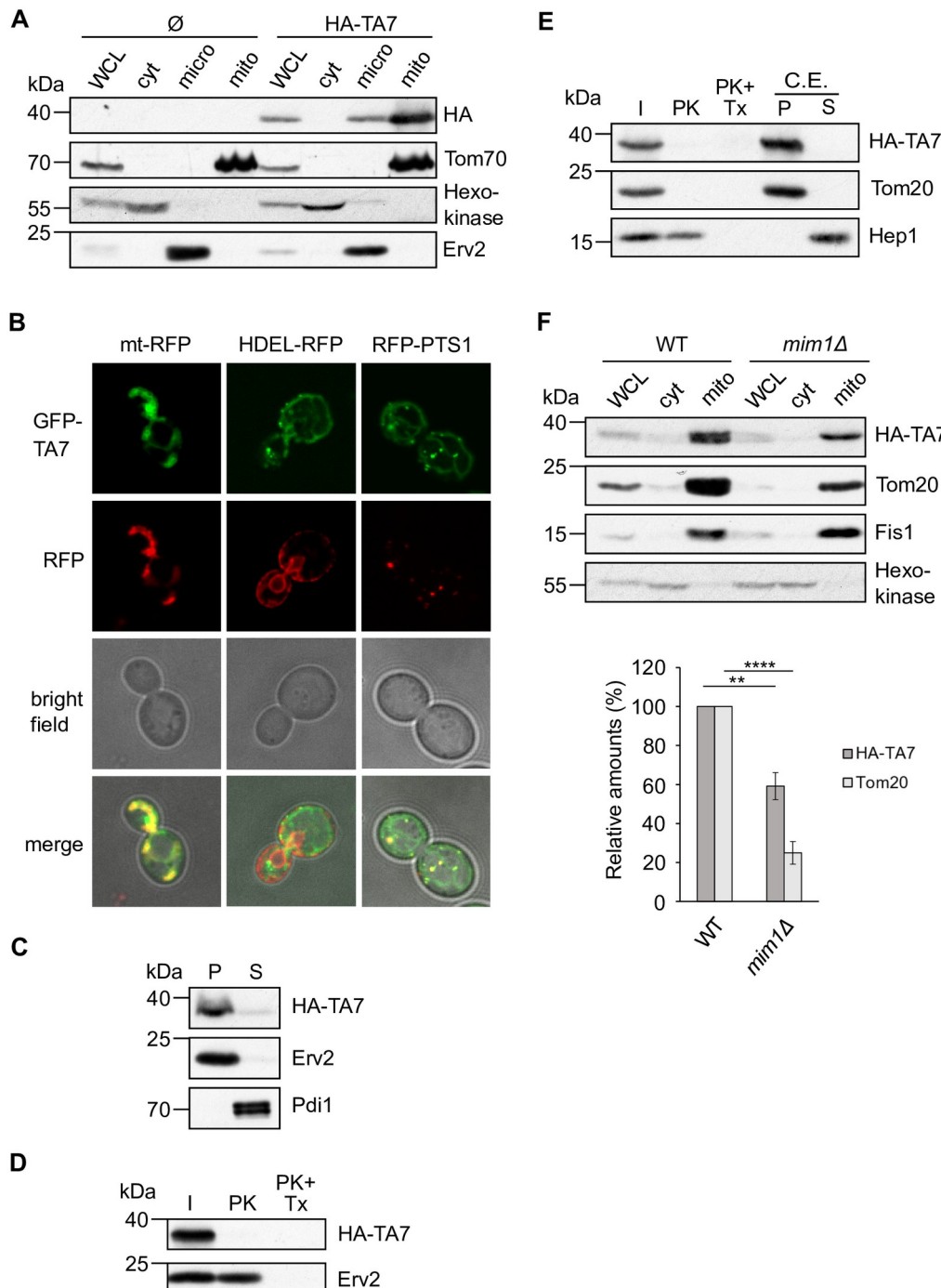

**Fig 2. TA7 expressed in yeast cells is targeted mainly to mitochondria.** (A) Subcellular fractionation of WT cells transformed with either an empty plasmid (Ø) or a plasmid encoding 3HA-TA7 was performed as described in Fig 1A. Tom70 and Hexokinase are mitochondrial and cytosolic proteins, respectively. (B) Representative fluorescence microscopy images of WT cells co-transformed with plasmids encoding GFP-TA7 and mt-RFP, HDEL-RFP, or RFP-PTS1. (C) Alkaline extraction of microsomal fractions isolated from WT cells expressing 3HA-TA7 was performed and analysed as described in Fig 1D. (D) Microsomes isolated from WT cells expressing 3HA-TA7 were treated and analysed as described in Fig 1E. (E) Mitochondria isolated as in (A) were treated and analysed as described in Fig 1G. Hep1 is a soluble matrix protein. (F) Top panel. Whole cell lysate (WCL), cytosol (cyt) and crude mitochondria (mito) were isolated from WT and *mim1Δ* cells expressing 3HA-TA7 and analysed by Western blot and immunodecoration. Bottom panel. The intensity of the bands corresponding to 3HA-TA7 and Tom20 in crude mitochondria from at least three independent experiments as in the top panel were quantified and normalized with PonceauS staining. The values are presented as percentage of their levels in WT cells. Error bars represent ±SD. ** $P \leq 0.01$; **** $P \leq 0.0001$.

**Table 5. Physicochemical features of hydrogenosomal and mitochondrial TA proteins.**

| | Hydrogenosomal proteins | | | Mitochondrial proteins | |
|---|---|---|---|---|---|
| | **TA4** | **TA7** | **TA10** | **Fis1** | **Gem1** |
| **Sequence TMS[a]** | FIGTMVAIGVGA, GLATHWLI | GISKPLIVGGAV, IAAGFLLYKGI | FYNKFWGIFSVV, AFFGVIIF | GVVVAGGVLAGA, VAVASFFL | TALIFGSTVGFV, ALCSFTLMKLF |
| **Position of TMS (a.a.)[a]** | 403–422 | 293–315 | 162–181 | 131–150 | 633–655 |
| **TMS length (a. a.)[a]** | 20 | 23 | 20 | 20 | 23 |
| **GRAVY of TMS[b]** | 1.63 | 1.41 | 1.68 | 2.25 | 1.69 |
| **Agadir score of TMS[c1]** | 0.23 | 0.58 | 0.24 | 0.30 | 0.40 |
| **Agadir score of TMS[c2]** | 0.21 | 0.54 | 0.22 | 0.27 | 0.35 |
| **Sequence CTE[d]** | KLITKVAKKKE | KRFMNHK | CKDEKK | RNKRR | KSSKFSK |
| **CTE length (a. a.)[d]** | 11 | 7 | 6 | 5 | 7 |
| **Net charge of CTE[d]** | +4 | +3 | +1 | +4 | +3 |
| **Sequence NTE (10 a.a.)[e]** | YGEPKPTDWK | HGIPFNLSKV | NEQNQEKLNS | EDKIQKETLK | TAAKDVDYRQ |
| **Net charge of NTE[e]** | 0 | +1 | -1 | 0 | 0 |

[a], TMS predicted according to TMHMM Server v. 2.0 (www.cbs.dtu.dk/services/TMHMM/)

[b], GRAVY, grand average of hydropathy (www.gravy-calculator.de)

[c1], Agadir score, % helical content was calculated at pH 7.5, 298 K, ionic strength 0.15 M using the Agadir prediction algorithm (http://agadir.crg.es) [24–27]

[c2], Agadir score, % helical content was calculated at pH 7.3, 303 K, ionic strength 0.15 M using the Agadir prediction algorithm (http://agadir.crg.es) [24–27]

[d], CTE, C-terminal element

[e], NTE, N-terminal element

was sensitive to PK, as the control protein Tom20, while the matrix protein Hep1 was protected from the protease, unless Triton-X was added (Fig 2E). Moreover, HA-TA7 was found in the pellet fraction after alkaline extraction, suggesting that it is a membrane-embedded protein (Fig 2E). In conclusion, TA7 behaves in yeast cells as a classical mitochondrial TA protein, suggesting that its targeting signal resembles that of a yeast mitochondrial TA protein. Of note, this signal is not highly specific, as a small fraction of TA7 is targeted also to both ER and peroxisomes.

To further investigate the insertion of TA7 into the MOM, we isolated crude mitochondria from WT and *mim1Δ* cells expressing HA-TA7 and analysed the steady state levels of the protein. Interestingly, we could observe a significant reduction (about 40%) of the amounts of HA-TA7 in the deletion strain, suggesting a role for the MIM complex in the insertion of TA7 into the MOM (Fig 2F). This finding substantiates a role for the MIM complex in the biogenesis of some mitochondrial TA proteins and its ability to also recognize *T. vaginalis* proteins. This result further suggests that, although the MIM complex is conserved only in fungi, it is able to recognize and process substrates from organisms that do not contain a homologous machinery. Despite recent efforts to identify the hydrogenosomal protein import machineries, no MIM complex has been identified yet [2, 7]. Therefore, most likely a protein or a protein complex, performing the same functions of the MIM complex evolved independently in *T. vaginalis*, as a result of convergent evolution. Support for this hypothesis came from our recent findings that the parasitic kinetoplastid *T. brucei* has a functional MIM orthologue,

pATOM36, which does not share any sequence similarities with the two subunits of the MIM complex [28].

## TA10 is equally distributed between ER and mitochondria

The TMS of TA10 has similar hydrophobicity, helical content, and length as that of TA4, although it has a shorter and less charged CTE (Table 5). Hence, we were interested to study whether these proteins are targeted to the same organelles in yeast. The subcellular fractionation of cells expressing HA-TA10 revealed that this protein is equally distributed between ER and mitochondria (Fig 3A). Of note, the mitochondrial protein Por1 and the ER protein Erv2, which served as control marker proteins, were detected only in their respective compartment. Surprisingly, we observed that the signal corresponding to HA-TA10 migrated at an apparent molecular weight of approximately 38 kDa although the expected one was 25 kDa. We re-analysed our construct and currently cannot explain this difference. The localisation of TA10 in both ER and mitochondria was further confirmed by fluorescence microscopy where co-localization of GFP-TA10 with both mitochondrial and ER markers was observed (Fig 3B). Of note, this hydrogenosomal protein did not show any localization to peroxisomes (Fig 3B).

Next, we investigated the topology of HA-TA10 in the ER membrane and observed that the protein is membrane-embedded and protected from the PK treatment, suggesting that the N-terminal domain is facing the lumen of the organelle (Fig 3C). Moreover, we noticed that the protein molecules localized in the ER had an apparent molecular weight higher than those detected in mitochondria (Fig 3A), suggesting a posttranslational modification in the ER. Indeed, we could demonstrate that HA-TA10 is glycosylated in the ER in yeast. Treatment with either EndoH or PNGase resulted in a migration shift, comparable to the one observed for the control glycosylated protein Pdi1 (Fig 3D). Given that the N-terminus of the protein is facing the lumen of the ER (Fig 3C), it is likely that this large soluble domain is glycosylated. However, the analysis of the sequence of TA10 with NetNGlyc 1.0 Server did not reveal any predicted glycosylation site.

Surprisingly, PK treatment of the mitochondrial fraction caused a reduction but not a complete loss of the HA-TA10 signal (Fig 3E). Additionally, the protein was detected in both the pellet and the supernatant fractions after alkaline extraction (Fig 3E). These findings suggest that about half of the mitochondrial-associated HA-TA10 molecules were not completely embedded in the membrane and that a small portion was inserted with a conformation that allows the protein to be protease-resistant due to either folding or a topology with the soluble domain facing the IMS. Taken together, these findings indicate that, similarly to TA4, TA10 can be targeted in yeast to both ER and mitochondria, where it can acquire an inverted topology.

We then investigated the effect of the loss of the MIM complex on the targeting of this hydrogenosomal protein to mitochondria. Interestingly, we could not observe a significant reduction of the levels of TA10 in crude mitochondria isolated from *mim1Δ* strain (Fig 3F), suggesting that the fraction of the protein that is inserted into the MOM, follow a MIM-independent pathway.

## The TMSs of the hydrogenosomal proteins can functionally replace the TMS of the mitochondrial TA protein Gem1

To confirm the potential of the hydrogenosomal proteins to correctly insert into the MOM, we tested the capacity of their TMSs to functionally replace the TMS of the mitochondrial TA protein Gem1. We investigated by serial dilutions the ability of these constructs to rescue the growth defect of a strain deleted for *GEM1*. WT or *gem1Δ* cells transformed with an empty

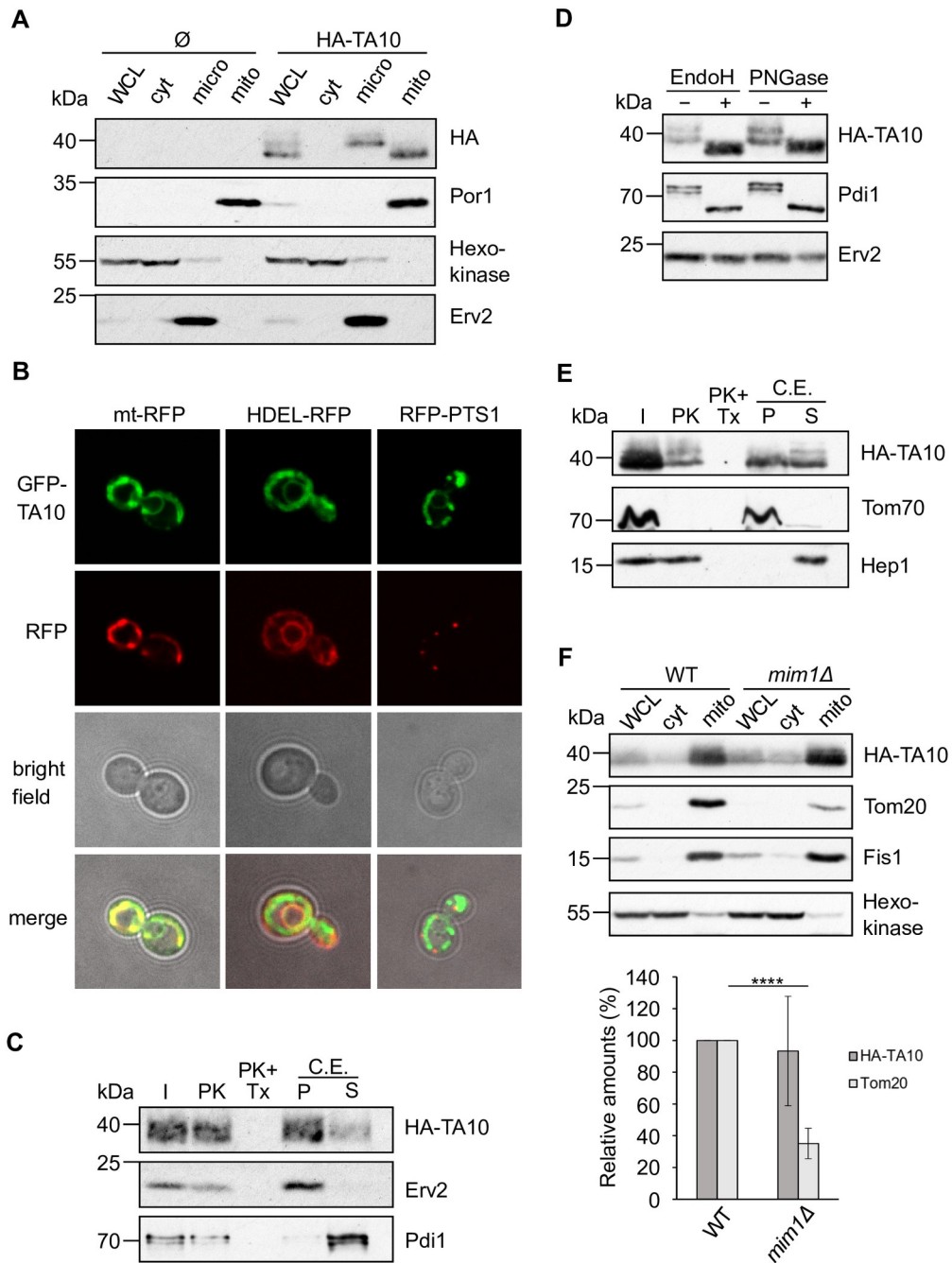

**Fig 3. TA10 is localized partially in the ER and is glycosylated there.** (A) Subcellular fractionation from WT cells transformed with either an empty plasmid (Ø) or a plasmid expressing 3HA-TA10 was performed and analysed as described in Fig 1A. (B) Representative fluorescence microscopy images of WT cells co-transformed with plasmids encoding GFP-TA10 and mt-RFP, HDEL-RFP, or RFP-PTS1. (C) PK treatment and alkaline extraction of microsomal fractions isolated from WT cells expressing 3HA-TA10. Samples were analysed by immunodecoration with the indicated antibodies. (D) The microsomal fractions isolated from cells expressing 3HA-TA10 were treated as in Fig 1F. (E) Mitochondria isolated as in (A) were treated and analysed as described in Fig 1G. Tom70 is a MOM protein facing the cytosol. (F) Top panel. WCL, cytosol (cyt) and crude mitochondria (mito) fractions were isolated from WT and *mim1Δ* cells expressing 3HA-TA10 and analysed by Western blot. Bottom panel. The intensity of the bands corresponding to 3HA-TA10 and Tom20 in crude mitochondria from at least three independent experiments as in the top panel were quantified and normalized with PonceauS. The levels are presented as percentage of their amounts in WT cells and error bars represent ±SD. **** P ≤ 0.0001.

plasmid, a vector encoding native Gem1, or the hybrid proteins, were spotted on a medium containing glycerol as carbon source, a condition that requires functional mitochondria. As expected, *gem1Δ* cells displayed a growth defect on non-fermentable carbon source at higher temperatures (Fig 4A, SG-Ura, 37°C) [29]. Of note, the hybrid proteins containing the TMS from the hydrogenosomal TA proteins rescued this phenotype as well as a plasmid-encoded

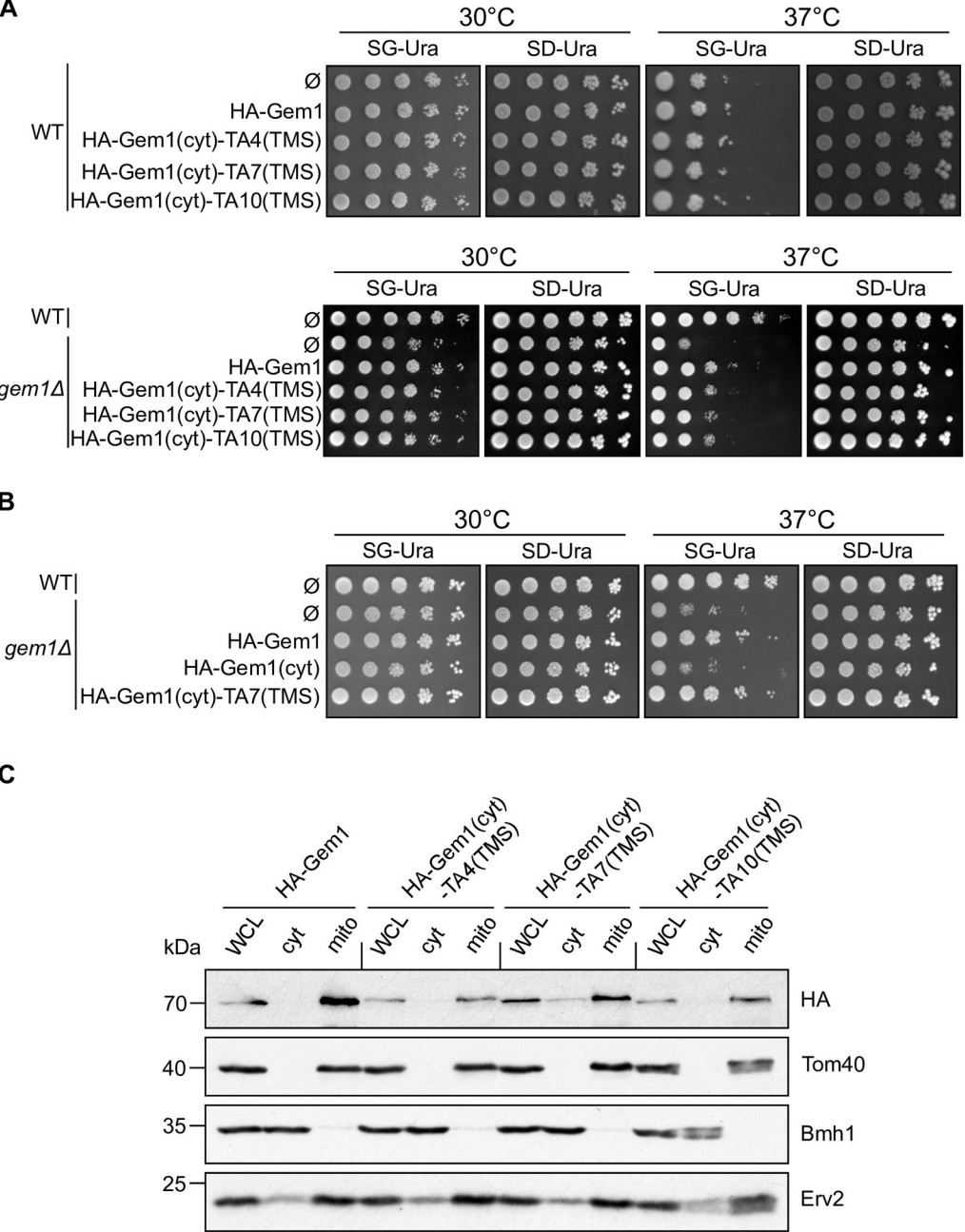

**Fig 4. Hybrid proteins that contain the TMS of hydrogenosomal proteins can rescue the absence of Gem1.** (A, B) Serial dilutions of the indicated strains were spotted on plates containing synthetic medium lacking uracil with either glycerol (SG-Ura) or glucose (SD-Ura) as carbon source. The plates were incubated at either 30°C or 37°C. (C) Whole cell lysate (WCL), cytosol (cyt), and crude mitochondria (mito) fractions were isolated from cells expressing the indicated construct and analysed by Western blot and immunodecoration with the indicated antibodies.

native Gem1 (Fig 4A), suggesting that the TMS of those proteins is sufficient for the correct insertion of the hybrid protein into the MOM. To substantiate the contribution of the TMS for the complementation of the growth defect, we verified that a protein that contains only the cytosolic domain of Gem1 was unable to rescue the growth phenotype of *gem1Δ* cells (Fig 4B). Although the TMS-containing chimeras were detected at rather lower levels compared to Gem1, they were enriched in crude mitochondria, suggesting that the targeting of Gem1 was not altered by the exchange of the TMS (Fig 4C). Nevertheless, since the isolated crude mito-chondria contain ER contaminations, as shown by the presence of Erv2, we cannot test whether a portion of the chimeras was mis-localised to the ER.

It is rather surprising that the TMS of TA4 and TA10 could support the rescue of the loss of Gem1, since sub-populations of these proteins are either mis-localised to the ER (Figs 1A, 1C, 3A and 3B), or inserted into the MOM with the soluble domain mainly facing the IMS (Figs 1G and 3E), while Gem1 is a classical TA protein, exposed to the cytosol. This observation sug-gests that a small portion of the TMS of TA4 or TA10 correctly inserted into the MOM is suffi-cient to support function. These chimeras are expressed under the control of the promoter of Fis1, a protein about 20 fold more abundant than Gem1 [30]. Therefore, it is sufficient that a minor fraction of these chimeras inserts into MOM to provide physiological like amounts of Gem1 at the MOM. Alternatively, it might be possible that the soluble domain of Gem1 con-tributes to the canonical TA-like orientation of the hybrid protein.

## Conclusions

This work revealed that the targeting signal of hydrogenosomal TA proteins is not completely conserved in yeast. In fact, all the three tested proteins are targeted in *S. cerevisiae*, to different extents, to ER, mitochondria, and peroxisomes. This is surprising since their TMSs have a low helical content and a very moderate hydrophobicity, hallmarks of mitochondrial TA proteins, and their hydrophobicity is even lower than that of the TMSs of Fis1 and Gem1 (Table 5). Ele-vated hydrophobicity and high helical content of the TMS in turn are considered to be charac-teristics of ER tail-anchored proteins [10, 11]. The only protein behaving in yeast as a mitochondrial TA protein is TA7. Its TMS has very low hydrophobicity and its CTE has net positive charge. Surprisingly, the helical content of TA7 is relatively high, even compared to the yeast TA proteins. The low hydrophobicity coupled with moderate helical content and pos-itive CTE are a typical targeting signal for peroxisomal TA proteins. Indeed, TA7 is the only hydrogenosomal TA protein localised upon its expression in yeast also in peroxisomes. Given the absence of peroxisomes in *T. vaginalis* [31], it is possible that in this organism no specific protein targeting signal for avoiding peroxisomes, evolved. Therefore, in yeast cells, the target-ing information of TA7 could be ambiguous and be recognized both as a mitochondrial and a peroxisomal signal. Similarly to Gem1 [15, 17], TA7 is the only *T. vaginalis* protein requiring the MIM complex for its insertion. Since an orthologue of such a complex has not been identi-fied yet in hydrogenosomes, most probably a complex without sequence similarity to the Mim proteins mediates the integration of TA7 to the hydrogenosomes membrane. Nevertheless, the absence of the MIM complex only partially affects the mitochondrial fraction of the hydroge-nosomal protein, suggesting the presence of at least one additional redundant pathway, or the ability of these proteins to insert into the lipid bilayer in an unassisted manner, similar to Fis1.

It remains elusive why, despite their rather moderate hydrophobic TMS and low helical content, both TA4 and TA10 are not targeted exclusively to mitochondria. As shown in Table 5, the TMSs of these hydrogenosomal proteins do have similar length and hydrophobic-ity as those of the mitochondrial TA proteins Gem1. Likewise, also other parameters such as the charge and the length of the C- or N-terminal elements are not much different from those

of the mitochondrial proteins. Of note, the apparent mistargeting of these hydrogenosomal proteins to the ER is in line with previous observations regarding the targeting of mammalian TA proteins expressed in yeast cells. For example, the cytochrome b5-RR variant, which is solely mitochondrial in mammal cells, was found also in the ER fraction upon its expression in yeast cells [22]. Similar difference between targeting in mammalian cells to that in yeast were also reported for protein tyrosine phosphatase (PTP)1B and Bcl-2. Both proteins are localized to the ER and mitochondria in mammalian cells but were detected, once expressed in yeast cells, exclusively in the ER [32, 33]. Thus, it appears that those traits that secure proper targeting in mammalian cells do not function correctly in yeast cells.

In summary, our results underline that our understanding of the targeting signals, which dictate the final location of TA proteins, is still limited. It is clear that there is a general, widely studied, trend concerning the physicochemical features of signals directing various TA proteins to specific organelles [9–11]. However, it is also obvious that these characteristics are somewhat redundant and are not sufficient for a direct correlation to a single target organelle. Further studies are required to unravel the tight regulation of the specificity of TA proteins localization. It will be interesting to verify in future studies whether such variations are correlated with altered lipid composition of the outer membrane of hydrogenosomes as compared to the MOM.

## Supporting information

**S1 Raw images.**
(PDF)

## Acknowledgments

We thank E. Kracker for excellent technical assistance and Dr. Tachezy and Prof. Schuldiner for plasmids.

## Author Contributions

**Conceptualization:** Andrè Ferdigg, Doron Rapaport, Daniela G. Vitali.

**Data curation:** Andrè Ferdigg, Daniela G. Vitali.

**Formal analysis:** Daniela G. Vitali.

**Investigation:** Kai S. Dimmer, Daniela G. Vitali.

**Methodology:** Andrè Ferdigg, Daniela G. Vitali.

**Project administration:** Doron Rapaport.

**Resources:** Doron Rapaport.

**Writing – original draft:** Andrè Ferdigg, Doron Rapaport, Daniela G. Vitali.

**Writing – review & editing:** Kai S. Dimmer.

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
