## [Decision Letter · Decision Letter 0]

20 Apr 2020

PONE-D-20-08560

Hydrogenosomal tail-anchored proteins are targeted to both mitochondria and ER upon their expression in yeast cells

PLOS ONE

Dear Dr. Rapaport,

Thank you for submitting your manuscript to PLOS ONE. After careful consideration, we feel that it has merit but does not fully meet PLOS ONE’s publication criteria as it currently stands. Therefore, we invite you to submit a revised version of the manuscript that addresses the points raised during the review process.

specifically, major improvements are required following the comments made by Reviewer-1

We would appreciate receiving your revised manuscript by Jun 04 2020 11:59PM. To enhance the reproducibility of your results, we recommend that if applicable you deposit your laboratory protocols in protocols.io, where a protocol can be assigned its own identifier (DOI) such that it can be cited independently in the future. For instructions see: http://journals.plos.org/plosone/s/submission-guidelines#loc-laboratory-protocols

We look forward to receiving your revised manuscript.

Kind regards,

David Chau

Academic Editor

PLOS ONE

Journal Requirements:

Reviewers' comments:

Reviewer's Responses to Questions

**Comments to the Author**

1. Is the manuscript technically sound, and do the data support the conclusions?

Reviewer #1: Partly

Reviewer #2: Yes

2. Has the statistical analysis been performed appropriately and rigorously? 

Reviewer #1: No

Reviewer #2: Yes

3. Have the authors made all data underlying the findings in their manuscript fully available?

Reviewer #1: Yes

Reviewer #2: Yes

4. Is the manuscript presented in an intelligible fashion and written in standard English?

Reviewer #1: Yes

Reviewer #2: Yes

5. Review Comments to the Author

Reviewer #1: In this study from Ferdigg et al the authors over-expressed 3 TA proteins from Trichomonas vaginalis, which in this organism are targeted to hydrogenosomes, in S. cerevisiae and looked at targeting. The rational was to assess if these two organisms have comparable targeting systems/signals. They found that one protein targeted the mitochondria as expected whilst the other two showed some ER targeting. Based on the sequence properties of the known TA protein targeting signal (the TMS and surrounding region) this could not be explained.

This study presents the results of original research, experimental work is performed to an acceptable standard, in particular the subcellular fractionation is convincing. Experiments are sufficiently well described and conclusions presented in an appropriate fashion in standard English which meets research integrity standards.

The work makes a very minor contribution to the scientific record but based on PLOS One criteria is acceptable for publication with some changes, including explaining some data analysis and adding some simple quantification. I also suggest a few points to be clarified or explained which I think could improve the manuscript.

The main obvious point for improvement would be to increase the number of TA proteins tested, ideally testing all the known Trichomonas vaginalis TA proteins. Failing that can the authors justify why they chose these 3 particular TA proteins and what the evidence is that they are really targeted to hydrogenosomes in Trichomonas vaginalis? Are they just found in the proteomic study?

All TA proteins used in the study are over-expressed, this is known to cause saturation of targeting signals and indeed the authors point this out. If you over-express the mitochondrial only yeast TA proteins to the extent described here (Fis1 and Gem1) do you also see exclusively mitochondrial targeting? How should we interpret this over-expression data?

Can the authors provide more explanation on why they think two of the TA proteins are not acting as standard TA proteins and are instead facing the ER lumen. Why do you think this is the case and what other proteins show this phenomena? Could it be that they are in fact not TA proteins and have an additional TMS or similar?

In the experiments replacing the TMS of Gem1 with the TMS of the other TA proteins, the growth assay is not hugely convincing, complemented growth looks similar for all 3 TA proteins. Have you checked the targeting of these construct or indeed the targeting of the TMS of the 3 TA proteins alone fused to GFP? This might be different to the full-length protein and thus alter your interpretation. Would be good to have a control with Gem1 lacking the TMS or with an ER targeting signal for comparison and/or a liquid growth assay here to back up this minor growth alteration.

As you mention peroxisomal proteins, and note that these show moderate hydrophobicity and basic residues at the CTE, have you also checked for any peroxisomal targeting of these proteins as this looks possible based on the sequence in the table?

Are there any other properties you could look at in terms of the targeting signals which might explain the differences you observe? For example, as you know Rao et al., 2016 (eLife 5:e21301) found a role for other physicochemical properties of the TMS such as helical content (Agadir) score. Is this worth looking at? Also how are the TMS length etc defined in the table?

More generally, I think it would be useful if the authors could put this study into a wider context and explain why it is important and what we really learn from this work and where this might lead us to moving forward?

Specific points:

Why are you using different control proteins every time in the fractionation experiments (1A, 2A, 3A)? Also , not clear if the same mass of protein added for each fraction, please specify.

Images in 1B, 2B, 3B, are much too small, suggest increase the size. It would also be good to have an ER marker to show ER targeting where appropriate.

Statistical analysis of graphs in 1F,2F,3F, is lacking. I can only assume because it is not significant.

Fig 2F – still get some mitochondrial insertion in absence of Mim1 (albeit reduced) – how, what are the Mim1-independent insertion routes? Also still get some Tom20 insertion without Mim1, but bigger decrease than TA7 – any comment why?

Minor:

Line 37: organelle(S)

Line 43: Which components are conserved, is the ER import machinery the same?

Line 51: peroxisomal membranes

Reviewer #2: The manuscript „Hydrogenosomal tail-anchored proteins are targeted to both mitochondria and ER upon their expression in yeast cells“ submitted by Ferdigg et al. (PONE-D-20-08560) convincingly shows that TA-proteins from Trichomonas vaginalis can have a bimodal distribution when expressed in yeast (S. cerevisiae). In conclusion, the authors hypothesize that targeting mechanisms of TA proteins may have evolved independently in different organism clades leading to partially overlapping membrane insertions. In general, the manuscript is technically sound and the experiments are well described and discussed. However, the results described in the last part of the manuscript (as the authors also mention) are not always straight forward to interpret and somewhat weaken the scientific significance of the study. Therefore, I would suggest including a few additional control experiments into the manuscript before final publication.

1) Expression of Gem1 hybrid proteins with TA4 TMS replacements, are interestingly shown rescue growth inhibition in Gem1-deficient yeast strains (Fig. 4), suggesting a correct translocation of the Gem1-hybrid protein into yeast mitochondria. However, as the authors also mention, it remains possible, that only a minor portion of the overexpressed protein is correctly targeted and sufficient for the rescue. Therefore, it is impossible to conclude if the TMS quality has a major impact on the TA targeting in the yeast unless additional subcellular co-localization data (IF or fractionation) is presented.

2) With regard to the observations with the Gem1-TA4-TMS-hybrid protein mentioned in (1), the authors suggest that other parts of the protein’s aa sequence influence the targeting or insertion process into yeast mitochondria. To substantiate this hypothesis it would be straight-forward to additionally analyze hybrid Gem1-proteins which contain either the N- or C-terminal flanking regions of the TA4 protein.

3) As a final conclusion of their observations, the authors state that “the targeting signal of TA proteins is only partially conserved between hydrogenosomes and mitochondria” (abstract) and “our results improve our understanding of the evolution of hydrogenosomes and mitochondria suggesting that together with the degeneration of mitochondria to hydrogenosomes their TA proteins were adopted to new characteristics of their target organell”. With regard to the fact that the targeting of Trichomonas TA proteins was only analyzed in S. cerevisiae and previous reports describe a comparable mistargeting of mammalian mitochondrial TA proteins in the same species such a statement is too general and should be substantiated by analyzing the targeting of the respective Trichomonas TA proteins in at least a second eukaryotic species. Alternatively, the authors should interpret their findings a bit more cautiously.

4) Table 5 should include the amino acid sequences of the respective described protein parts (TMS, CTE NTE segments). It is quite tedious for the reader to having to sort out this information from the databases.

Minor points:

• Table 4: For the apparently self-made antibodies (source Lab stocks) the author should cite the reference in which the antibodies were initially characterized.

• Page 3, line 51: “peroxisome/peroxisomal membranes” not “peroxisomes membrane”

• Page 7: “the pellet (microsomes) ... was resuspended ..., homogenized and clarified by ...”. It is unclear how an organelle suspension was homogenized. Which technology was used?

• Page 10, line 227: “experiments” not “experimets”

• Page 11, line 248: “species” not “specie”

6. PLOS authors have the option to publish the peer review history of their article (what does this mean?). If published, this will include your full peer review and any attached files.

Reviewer #1: No

Reviewer #2: No

---

## [Author Response · Author response to Decision Letter 0]

20 Jul 2020

Addressing the comments of the reviewers

Reviewer #1:

We thank the reviewer for his/her generally positive opinion on our contribution.

The main obvious point for improvement would be to increase the number of TA proteins tested, ideally testing all the known Trichomonas vaginalis TA proteins. Failing that can the authors justify why they chose these 3 particular TA proteins and what the evidence is that they are really targeted to hydrogenosomes in Trichomonas vaginalis? Are they just found in the proteomic study?

A proteomic and bioinformatic study in Trichomonas vaginalis identified 12 putative hydrogenosomal TA proteins (Rada et al., PLOS ONE, 2011). However, in the same report and in a later study in T. vaginalis, the targeting and topology of only five of those proteins have been confirmed by immunofluorescence and biochemical approaches (Rada et al., Mol Microbiol, 2019). Since the hydrophobicity of the tail-anchor of mitochondrial TA proteins is usually lower than that of the tail-anchor of ER TA proteins, we chose for our study out of those five proteins the three that had the transmembrane segment with the lowest hydrophobicity.

All TA proteins used in the study are over-expressed, this is known to cause saturation of targeting signals and indeed the authors point this out. If you over-express the mitochondrial only yeast TA proteins to the extent described here (Fis1 and Gem1) do you also see exclusively mitochondrial targeting? How should we interpret this over-expression data?

The reviewer raised here a valid point. The tested TA proteins are recombinant proteins expressed in a heterologous organism. Since there is no endogenous level of those proteins in S. cerevisiae, it is difficult to know whether they are actually over-expressed. Nevertheless, those proteins were expressed under a strong promoter (TPIpr), therefore it is important to test whether such high protein levels could cause mis-localization. To address this point, we investigated the subcellular localization of the mitochondrial yeast TA protein Fis1 encoded by a plasmid resulting in expression under the control of the same promotor. The new results, which are included in the revised Fig. 1B, demonstrate that the protein is clearly localized exclusively to the mitochondrial fraction. Hence, we added to the revised text a sentence stating that these findings suggest that such high protein levels are not sufficient to trigger mistargeting of a mitochondrial TA protein to the ER. 

We could not perform similar experiments with overexpressed Gem1, because this protein in unstable and would be degraded during the long time required to perform a complete subcellular fractionation.

Can the authors provide more explanation on why they think two of the TA proteins are not acting as standard TA proteins and are instead facing the ER lumen. Why do you think this is the case and what other proteins show this phenomena? Could it be that they are in fact not TA proteins and have an additional TMS or similar?

These are indeed puzzling findings that we cannot explain. We can only speculate that this inverted topology is caused by differences in the lipid composition of the target membranes and/or variations in the way these proteins are processed by import factors. Since we do not have information on the lipid composition of hydrogenosomes or their import of tail-anchored proteins, these are pure speculations that we prefer not to include in the manuscript.

In a previous study, we observed that the mammalian cytochrome b5-RR variant, which is targeted in mammalian cells solely to mitochondria, was partially targeted to the ER with an inverted topology upon its expression in yeast (Vitali et al., J Cell Sci, 2018). The description of this previous observation was expended in the revised version. The topology of the hydrogenosomal TA has been investigated in both aforementioned publications (Rada et al., PLOS ONE, 2011, Rada et al., Mol Microbiol, 2019), and revealed that those proteins have the typical TA protein topology. Furthermore, employing different prediction programs, we and the two previous studies could not find in these proteins additional putative TMSs.

In the experiments replacing the TMS of Gem1 with the TMS of the other TA proteins, the growth assay is not hugely convincing, complemented growth looks similar for all 3 TA proteins. Have you checked the targeting of these construct or indeed the targeting of the TMS of the 3 TA proteins alone fused to GFP? This might be different to the full-length protein and thus alter your interpretation. Would be good to have a control with Gem1 lacking the TMS or with an ER targeting signal for comparison and/or a liquid growth assay here to back up this minor growth alteration.

As the reviewer suggested, we monitored the localization of the fusion proteins harbouring the cytosolic domain of Gem1. As shown in revised Fig. 4C, the constructs with Gem1 cytosolic domain fused to the TMS of the hydrogenosomal TA proteins are found in the crude mitochondria fractions. This observation, combined with the functional complementation, suggests that the TMS alone is sufficient for correct targeting, of at least a portion of the hybrid molecules, to mitochondria. However, since the isolated crude mitochondria fractions are contaminated by ER structures, it is not possible to certainly assess the percentage of the chimera proteins localized in mitochondria. 

We could not perform full subcellular fractionation experiments with these constructs because the presence of the cytosolic domain of Gem1 makes them unstable and we observed that they are degraded during the long time required to perform a subcellular fractionation.

We optimized our growth assay and revised Fig. 4A shows a clearer effect on cell growth at 37°C on non-fermentable carbon source upon deletion of GEM1. We agree with the reviewer that all three hybrid TA proteins complement the growth to the same extent. We explain this similar effect by the fact that these hybrid proteins were expressed under the control of FIS1 promoter. A recent proteomic study (Morgenstern et al., Cell Reports, 2017) reported that Fis1 is 20-fold more abundant than Gem1. Thus, even if only 5% of the molecules of each hybrid protein were correctly integrated into the mitochondrial OM, the number of Gem1 molecules anchored at the OM would resemble its normal amount.

In addition, according to the reviewer’s proposal, we have tested the ability of a construct containing only the cytosolic domain of Gem1 to rescue the gem1Δ strain growth phenotype. As expected and as shown in revised Fig. 4B, such a construct was unable to rescue the moderate growth retardation of gem1Δ cells.

Following the reviewer suggestion, we also performed a liquid growth assay. However, we could not detect a significant retardation of the gem1Δ strain under these conditions. Thus, this approach was not followed further.

As you mention peroxisomal proteins, and note that these show moderate hydrophobicity and basic residues at the CTE, have you also checked for any peroxisomal targeting of these proteins as this looks possible based on the sequence in the table?

Following the comment of the reviewer, we investigated the targeting of the GFP-tagged hydrogenosomal TA proteins to peroxisomes by fluorescence microscopy using RFP tagged with PTS1 (RFP-PTS1) as a peroxisomal marker. GFP-TA4 and GFP-TA10 did not show any colocalization with peroxisomes (revised Figs. 1C and 3B), while GFP-TA7 was targeted, in addition to mitochondria, also to peroxisomes (revised Fig. 2B). We explain this observation by the higher helical content of the TMS of TA7 as compared to those of the other TA proteins (revised Table 5). Relatively high helical content is a feature that is typical for peroxisomal TA proteins. These new results are described and discussed in the revised text.

Are there any other properties you could look at in terms of the targeting signals which might explain the differences you observe? For example, as you know Rao et al., 2016 (eLife 5:e21301) found a role for other physicochemical properties of the TMS such as helical content (Agadir) score. Is this worth looking at? Also how are the TMS length etc defined in the table?

We followed the suggestion of the reviewer and in the revised Table 5 we included the helical content (Agadir) score, calculated both in the same conditions used by Rao et al., eLife, 2016 (pH 7.5, 25 °C, 0.15 M) and adjusted to yeast physiological conditions (pH 7.3, 30°C, 0.15 M). TA7 has a higher helical content than the yeast TA proteins, which is in line with its partial localization to peroxisomes. However, the very low helical contents of TA4 and TA10 is in contrast with their partial localization to the ER, which on average requires a higher helical content and a more hydrophobic TMS. Thus, it seems that rather than a single physicochemical parameter, a combination of several factors dictate the subcellular distribution of TA proteins. This point is discussed in the revised text.

Additionally, in the revised Table 5 we added details on how each parameter was defined.

More generally, I think it would be useful if the authors could put this study into a wider context and explain why it is important and what we really learn from this work and where this might lead us to moving forward?

In the Conclusion section of revised text we elaborated on the impact of our study in a wider context.

Specific points:

Why are you using different control proteins every time in the fractionation experiments (1A, 2A, 3A)? Also , not clear if the same mass of protein added for each fraction, please specify.

The tested hydrogenosomal TA proteins have different molecular weight. For the immunodecoration with different antibodies we wanted to use strips from the very same gel. Hence, we had to choose the marker proteins such that all the used antibodies could be used with one gel. To allow a direct comparison of the protein intensities in the various fractions, we loaded in every experiment the same mass of each fraction on the gel. We included this information in the revised figure legends.

Images in 1B, 2B, 3B, are much too small, suggest increase the size. It would also be good to have an ER marker to show ER targeting where appropriate.

As requested, the revised Fig. 1C, 2B, and 3B are larger and include mitochondria, ER, and peroxisomal markers.

Statistical analysis of graphs in 1F,2F,3F, is lacking. I can only assume because it is not significant.

The reviewer’s assumption is correct. Statistical analysis has been included only in case where differences were statistically significant. Accordingly, statistical analysis was added to the graphs of revised Figs. 1H, 2F and 3F.

Fig 2F – still get some mitochondrial insertion in absence of Mim1 (albeit reduced) – how, what are the Mim1-independent insertion routes? Also still get some Tom20 insertion without Mim1, but bigger decrease than TA7 – any comment why?

The MIM complex is differentially required for the insertion of mitochondrial α-helical outer membrane proteins (Vitali et al., iScience, 2020 and Doan et al., Cell Rep. 2020). Accordingly, the loss of this import complex affects each substrate to a different extent. On one side of the spectrum, the mitochondrial TA protein Fis1 is inserted into the membrane in an unassisted, mainly MIM-independent, manner. On the other hand, Tom20 biogenesis is heavily dependent on the MIM complex and in the absence of the latter only a slow residual unassisted insertion of Tom20 can be observed. We propose that TA7 is in between these two examples; its membrane integration is normally facilitated by the MIM complex, but can take place also in an unassisted manner. We extended this point in the revised manuscript. 

Minor:

Line 37: organelle(S)

Line 43: Which components are conserved, is the ER import machinery the same?

Line 51: peroxisomal membranes

Lines 37 and 51: These minor comments have been adjusted in the revised manuscript.

Line 43: We modified our text. To our knowledge no information is available concerning the ER import machinery in Trichomonas vaginalis.

Reviewer #2: 

We thank the reviewer for his/her generally positive opinion on our contribution.

1) Expression of Gem1 hybrid proteins with TA4 TMS replacements, are interestingly shown rescue growth inhibition in Gem1-deficient yeast strains (Fig. 4), suggesting a correct translocation of the Gem1-hybrid protein into yeast mitochondria. However, as the authors also mention, it remains possible, that only a minor portion of the overexpressed protein is correctly targeted and sufficient for the rescue. Therefore, it is impossible to conclude if the TMS quality has a major impact on the TA targeting in the yeast unless additional subcellular co-localization data (IF or fractionation) is presented. 

A similar point was raised also by reviewer #1. As the reviewer suggested, we monitored the localization of the fusion proteins harbouring the cytosolic domain of Gem1. As shown in revised Fig. 4C, the constructs with Gem1 cytosolic domain fused to the TMS of the hydrogenosomal TA proteins are found in the crude mitochondria fractions. This observation, combined with the functional complementation, suggests that the TMS alone is sufficient for correct targeting, of at least a portion of the hybrid molecules to mitochondria. However, since the isolated crude mitochondria fractions are contaminated by ER structures, it is not possible to certainly assess the percentage of the chimera proteins localized in mitochondria. 

We could not perform full subcellular fractionation experiments with these constructs because the presence of the cytosolic domain of Gem1 makes them unstable and we observed that they are degraded during the long time required to perform a subcellular fractionation.

2) With regard to the observations with the Gem1-TA4-TMS-hybrid protein mentioned in (1), the authors suggest that other parts of the protein’s aa sequence influence the targeting or insertion process into yeast mitochondria. To substantiate this hypothesis it would be straight-forward to additionally analyze hybrid Gem1-proteins which contain either the N- or C-terminal flanking regions of the TA4 protein. 

We think that additional regions from the TA proteins might affect their membrane topology rather than targeting or membrane integration per se. The hybrid proteins analysed in Fig. 4 already contain the flanking regions of the TMS of the hydrogenosomal TA proteins. In the Materials and Methods section of the revised manuscript we included the corresponding protein sequence of each construct.

3) As a final conclusion of their observations, the authors state that “the targeting signal of TA proteins is only partially conserved between hydrogenosomes and mitochondria” (abstract) and “our results improve our understanding of the evolution of hydrogenosomes and mitochondria suggesting that together with the degeneration of mitochondria to hydrogenosomes their TA proteins were adopted to new characteristics of their target organell”. With regard to the fact that the targeting of Trichomonas TA proteins was only analyzed in S. cerevisiae and previous reports describe a comparable mistargeting of mammalian mitochondrial TA proteins in the same species such a statement is too general and should be substantiated by analyzing the targeting of the respective Trichomonas TA proteins in at least a second eukaryotic species. Alternatively, the authors should interpret their findings a bit more cautiously. 

We followed the recommendation of the reviewer and modified our statements in the Abstract and Conclusions sections of the revised manuscript to present more cautious interpretations.

4) Table 5 should include the amino acid sequences of the respective described protein parts (TMS, CTE NTE segments). It is quite tedious for the reader to having to sort out this information from the databases.

As suggested, we have included the amino acid sequences of TMSs, CTEs, and NTEs in the revised Table 5.

Minor points:

• Table 4: For the apparently self-made antibodies (source Lab stocks) the author should cite the reference in which the antibodies were initially characterized.

Most of our antibodies are not commercially available and have not been generated in our laboratory. Instead, they have been collected for many years from different laboratories in the community. Although we cannot track the very first time these antibodies were characterized, all of them were used in many publications.

• Page 3, line 51: “peroxisome/peroxisomal membranes” not “peroxisomes membrane”

• Page 7: “the pellet (microsomes) ... was resuspended ..., homogenized and clarified by ...”. It is unclear how an organelle suspension was homogenized. Which technology was used?

• Page 10, line 227: “experiments” not “experimets”

• Page 11, line 248: “species” not “specie”

Page 3, 10, and 11: All these minor comments were addressed in the revised manuscript. 

Page 7: The microsomes pellet was homogenized with a douncer, as described in the revised Materials and Methods section.

---

## [Decision Letter · Decision Letter 1]

4 Aug 2020

PONE-D-20-08560R1

Hydrogenosomal tail-anchored proteins are targeted to both mitochondria and ER upon their expression in yeast cells

PLOS ONE

Dear Dr. Vitali,

Thank you for submitting your manuscript to PLOS ONE. After careful consideration, we feel that it has merit but does not fully meet PLOS ONE’s publication criteria as it currently stands. Therefore, we invite you to submit a revised version of the manuscript that addresses the points raised during the review process.

We look forward to receiving your revised manuscript.

Kind regards,

David Chau

Academic Editor

PLOS ONE

Reviewers' comments:

Reviewer's Responses to Questions

**Comments to the Author**

1. If the authors have adequately addressed your comments raised in a previous round of review and you feel that this manuscript is now acceptable for publication, you may indicate that here to bypass the “Comments to the Author” section, enter your conflict of interest statement in the “Confidential to Editor” section, and submit your "Accept" recommendation.

Reviewer #1: All comments have been addressed

Reviewer #2: (No Response)

2. Is the manuscript technically sound, and do the data support the conclusions?

Reviewer #1: Yes

Reviewer #2: Yes

3. Has the statistical analysis been performed appropriately and rigorously? 

Reviewer #1: Yes

Reviewer #2: Yes

4. Have the authors made all data underlying the findings in their manuscript fully available?

Reviewer #1: Yes

Reviewer #2: Yes

5. Is the manuscript presented in an intelligible fashion and written in standard English?

Reviewer #1: Yes

Reviewer #2: Yes

6. Review Comments to the Author

Reviewer #1: (No Response)

Reviewer #2: The revised version of the manuscript PONE-D-20-08560R1 has improved substantially and the reviewer requests were generally properly fulfilled. There is just one last point of criticism regarding a conclusions of this study. On page 26, line 529-531 the authors conclude that their "study suggests that TA proteins might have evolved after mitochondria became hydrogenosomes". With regard to the results presented, such an assumption is not justified. As the authors cite in the manuscript, also mammalian mitochondrial TA-proteins have been found to mistarget to the ER in the same fashion as the Trichomonas TA proteins investigated. Thus, also mammals have developed a TA targeting system which differs substantially from the yeast system. However, they are evolutionary closer related to yeast than Trichomonas and contain normal mitochondria. In the light of these discrepancies the sentence should be removed from the manuscript.

7. PLOS authors have the option to publish the peer review history of their article (what does this mean?). If published, this will include your full peer review and any attached files.

Reviewer #1: No

Reviewer #2: **Yes: **Markus Islinger

---

## [Author Response · Author response to Decision Letter 1]

4 Aug 2020

Addressing the comments of the reviewers

Reviewer #2:

We thank the reviewer for his/her generally positive opinion on our contribution.

The revised version of the manuscript PONE-D-20-08560R1 has improved substantially and the reviewer requests were generally properly fulfilled. There is just one last point of criticism regarding a conclusions of this study. On page 26, line 529-531 the authors conclude that their "study suggests that TA proteins might have evolved after mitochondria became hydrogenosomes". With regard to the results presented, such an assumption is not justified. As the authors cite in the manuscript, also mammalian mitochondrial TA-proteins have been found to mistarget to the ER in the same fashion as the Trichomonas TA proteins investigated. Thus, also mammals have developed a TA targeting system which differs substantially from the yeast system. However, they are evolutionary closer related to yeast than Trichomonas and contain normal mitochondria. In the light of these discrepancies the sentence should be removed from the manuscript.

The reviewer raised a valid point and we agree that the conclusion that “TA proteins might have evolved after mitochondria became hydrogenosomes” is merely a speculation. Therefore, we removed that sentence from the revised manuscript.

---

## [Decision Letter · Decision Letter 2]

7 Aug 2020

Hydrogenosomal tail-anchored proteins are targeted to both mitochondria and ER upon their expression in yeast cells

PONE-D-20-08560R2

Dear Dr. Vitali,

We’re pleased to inform you that your manuscript has been judged scientifically suitable for publication and will be formally accepted for publication once it meets all outstanding technical requirements.

Kind regards,

David Chau

Academic Editor

PLOS ONE

Additional Editor Comments (optional):

Reviewers' comments:

Reviewer's Responses to Questions

**Comments to the Author**

1. If the authors have adequately addressed your comments raised in a previous round of review and you feel that this manuscript is now acceptable for publication, you may indicate that here to bypass the “Comments to the Author” section, enter your conflict of interest statement in the “Confidential to Editor” section, and submit your "Accept" recommendation.

Reviewer #2: All comments have been addressed

2. Is the manuscript technically sound, and do the data support the conclusions?

Reviewer #2: Yes

3. Has the statistical analysis been performed appropriately and rigorously? 

Reviewer #2: Yes

4. Have the authors made all data underlying the findings in their manuscript fully available?

Reviewer #2: Yes

5. Is the manuscript presented in an intelligible fashion and written in standard English?

Reviewer #2: Yes

6. Review Comments to the Author

Reviewer #2: (No Response)

7. PLOS authors have the option to publish the peer review history of their article (what does this mean?). If published, this will include your full peer review and any attached files.

Reviewer #2: **Yes: **Markus Islinger

---

## [Editor Report · Acceptance letter]

10 Aug 2020

PONE-D-20-08560R2 

Hydrogenosomal tail-anchored proteins are targeted to both mitochondria and ER upon their expression in yeast cells 

Dear Dr. Vitali:

I'm pleased to inform you that your manuscript has been deemed suitable for publication in PLOS ONE. Congratulations! Your manuscript is now with our production department. 

Kind regards, 

on behalf of

Dr. David Chau 

Academic Editor

PLOS ONE